# Validation of a pre-established triage protocol for critically ill patients in a COVID-19 outbreak under resource scarcity: A retrospective multicenter cohort study

**Nicolas Donat**[1]*, **Nouchan Mellati**[2,3], **Thibault Frumento**[4], **Audrey Cirodde**[1],
**Sébastien Gette**[2], **Pierre Gildas Guitard**[4], **Clément Hoffmann**[1], **Benoît Veber**[4,5],
**Thomas Leclerc**[1,6]

**1** Burn Treatment Center and COVID-19 ICU, Percy Military Teaching Hospital, Clamart, France, **2** ICU, Mercy Regional Hospital, Metz, France, **3** Legouest Military Teaching Hospital, Metz, France, **4** ICU, Rouen University Hospital, Rouen, France, **5** Faculty of Medicine, Rouen University, Rouen, France, **6** Val-de-Grâce Military Medical Academy, Paris, France

* nicolas.donat@intradef.gouv.fr

## Abstract

### Introduction

In case of COVID-19 related scarcity of critical care resources, an early French triage algorithm categorized critically ill patients by probability of survival based on medical history and severity, with four priority levels for initiation or continuation of critical care: P1 –high priority, P2 –intermediate priority, P3 –not needed, P4 –not appropriate. This retrospective multicenter study aimed to assess its classification performance and its ability to help saving lives under capacity saturation.

### Methods

ICU patients admitted for severe COVID-19 without triage in spring 2020 were retrospectively included from three hospitals. Demographic data, medical history and severity items were collected. Priority levels were retrospectively allocated at ICU admission and on ICU day 7–10. Mortality rate, cumulative incidence of death and of alive ICU discharge, length of ICU stay and of mechanical ventilation were compared between priority levels. Calculated mortality and survival were compared between full simulated triage and no triage.

### Results

225 patients were included, aged 63.1±11.9 years. Median SAPS2 was 40 (IQR 29–49). At the end of follow-up, 61 (27%) had died, 26 were still in ICU, and 138 had been discharged. Following retrospective initial priority allocation, mortality rate was 53% among P4 patients (95CI 34–72%) versus 23% among all P1 to P3 patients (95CI 17–30%, chi-squared p = 5.2e-4). The cumulative incidence of death consistently increased in the order P3, P1, P2 and P4 both at admission (Gray's test p = 3.1e-5) and at reassessment (p = 8e-5), and conversely for that of alive ICU discharge. Reassessment strengthened consistency. Simulation

**Data Availability Statement:** All relevant data are within the paper and its Supporting information files.

**Funding:** The authors received no specific funding for this work.

**Competing interests:** The authors have declared that no competing interests exist.

under saturation showed that this two-step triage protocol could have saved 28 to 40 more lives than no triage.

## Conclusion

Although it cannot eliminate potentially avoidable deaths, this triage protocol proved able to adequately prioritize critical care for patients with highest probability of survival, hence to save more lives if applied.

## Introduction

The worldwide spread of SARS-Cov-2 disease, or COVID-19, since the end of 2019 has resulted in iterative surges of patients with severe respiratory failure requiring critical care, causing tension or saturation of critical care capacities all over the world at various periods [1–3]. First responses associated various mixes of capacity increase and transmission control, the former by redistributing hospital resources, opening new temporary intensive care unit (ICU) beds and evacuating patients to relatively spared areas, and the latter based on barrier prevention measures, testing, contact tracing, isolation of cases and contacts, and various types of population lock-down [4]. From 2021 on, newly developed vaccines proved highly efficient against severe COVID-19 [5, 6]. This has efficiently reduced the strain on ICUs, at least in areas where mass vaccination has led to a high immunization rate [7].

COVID-19 has been dramatically illustrating the challenges of emerging infectious diseases when their severity and contagiousness strain acute care capacities to their limits. Considering the finitude of human and logistical resources, even with temporary capacity increases and until immunity is reached, resources can become too scarce to meet requirements, especially for highly demanding activities such as critical care [8, 9]. Triage to prioritize critical care initiation and continuation for patients who have the highest probability of benefiting from treatment then becomes an ethical necessity to save the greatest number of lives [10–14].

However, early triage recommendations during the COVID-19 pandemic kept to general principles, leaving physicians with limited practical guidance and risks of misjudgment. In order to fill this gap, building upon experience of disaster and war medicine and inspiring from a Canadian influenza triage scheme, from Swiss guidelines and from other COVID-19 specific considerations, the French Society of Anesthesia and Critical Care Medicine (Société Française d'Anesthésie et de Réanimation, SFAR) and the French Military Medical Service (Service de Santé des Armées, SSA) issued critical care triage guidelines under COVID-19 related resource scarcity during the first epidemic wave [10, 13–17]. The original SFAR/SSA guidelines were made available online on April 4th, 2020 on the SFAR website, and their English translation published in June 2020 [17].

Briefly, the SFAR/SSA critical care triage protocol distinguishes between two types of crisis situations, tension and saturation. *Tension* describes situations where critical care can be provided to all eligible patients only with major efforts to extend ICU capacity and to transfer patients. *Saturation* describes situations when, even with such efforts, the limited available resources only allow to provide critical care to part of eligible patients. In both situations, patients are triaged in four priority levels for critical care initiation or continuation according to medical history and actual severity, and based on the anticipated probability that critical care enables a favorable outcome in reasonable time: P1 –high priority; P2 –intermediate priority; P3 –low priority, critical care not, not yet or no longer needed; P4 –last priority due to

poorest anticipated outcome. Initial triage is reassessed at typical disease turning point (day 7 to 10 for COVID-19) or when the availability of resources changes [17]. Figs 1 and 2 summarize this triage algorithm.

In order to estimate both the adequacy and usability of the SFAR/SSA critical care triage protocol in epidemic waves of COVID-19 without sufficient population immunity, its ability to classify patients based on their probability of survival under critical care treatment was retrospectively assessed in a cohort of COVID-19 ICU patients during the first epidemic wave in France.

## Methods

### Study design and setting

This retrospective multi-center cohort study received ethical approval from the institutional review board (CERAR, ethical committee for research in anesthesia and critical care medicine) of the French Society of Anesthesia and Critical Care Medicine (SFAR), who waived the requirement of informed consent (reference IRB 00010254–2020–090), and by local institutional review boards when required. The STROBE recommendations for observational studies were followed [18].

### Setting

The study was conducted in the COVID-19 extended ICUs of three separate hospitals: one regional hospital, one military teaching hospital and one university hospital. Before COVID-19, the French healthcare system routinely offered 5400 to 5500 ICU beds, 1225 of them located in Paris area (Île-de-France region) [19, 20]. During the first epidemic wave of COVID-19, new ICU beds were created from acute care units, post-anesthesia care units and operating theaters [19]. Overall ICU capacity in France was progressively raised to 10700 beds, including 2700 in Île-de-France, based on records from the French Ministry of Health [1]. At the peak of the outbreak on April 8th, 2020, the overall ICU capacity in France had been increased by 95% and 7148 COVID-19 patients were in ICU [19]. Owing to this capacity surge and to many inter-region transfers, no systematized triage was used for the initiation and continuation of critical care at that time.

### Patients

Adult patients diagnosed with COVID-19, either based on positive SARS-CoV-2 RT-PCR from a nasal or oropharyngeal swab or from tracheal of lower respiratory tract sampling, or based on high clinical suspicion with suggestive medical history and typical CT-scan findings, and admitted to ICUs of participating hospitals between March and May 2020 were retrospectively included. Patients lost to follow-up, *i.e.* with unknown status at the time of data collection due to transfer to other facilities, were excluded.

### Collected data

All data were retrospectively collected from patient medical records. Demographic data, past medical history (using closed questions) and chronological evolution of COVID-19 (symptom onset, diagnosis and ICU admission dates) were recorded. Organ failure and severity criteria were recorded on arrival then again after 7 to 10 ICU days according to the SFAR/SSA prioritization protocol, using the latest date with highest data completeness within that range. Following items were collected: sequential organ failure assessment (SOFA) score components, namely PaO2/FiO2 ratio and need for mechanical ventilation, mean arterial pressure (MAP)

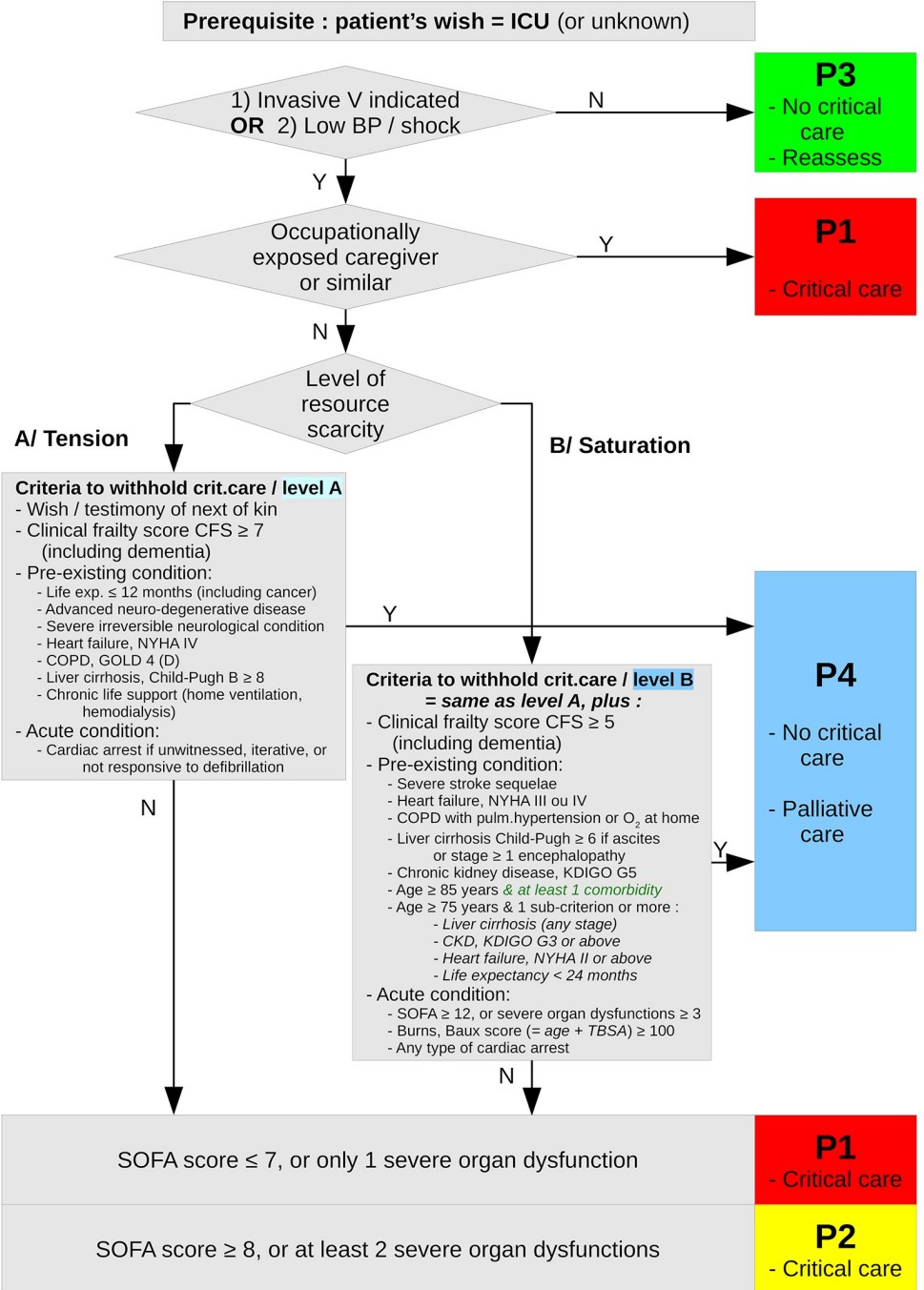

**Fig 1. Triage algorithm for critical care initiation under resource scarcity due to COVID-19.** Summary of the first step (day 0, critical care initiation) of the SFAR/SSA critical care prioritization/triage protocol, adapted from [17] with proposed substitution of "age ≥ 85 & at least one comorbidity" to "age ≥ 85" alone.

and catecholamine infusion rate, Glasgow coma scale, serum creatinine, platelet count and serum bilirubin, along with simplified acute physiology score (SAPS2), extra-corporeal membrane oxygenation (ECMO), and cardiac arrest occurrence. Mechanical ventilation was defined as either invasive or non-invasive, but not high flow nasal oxygen although it was

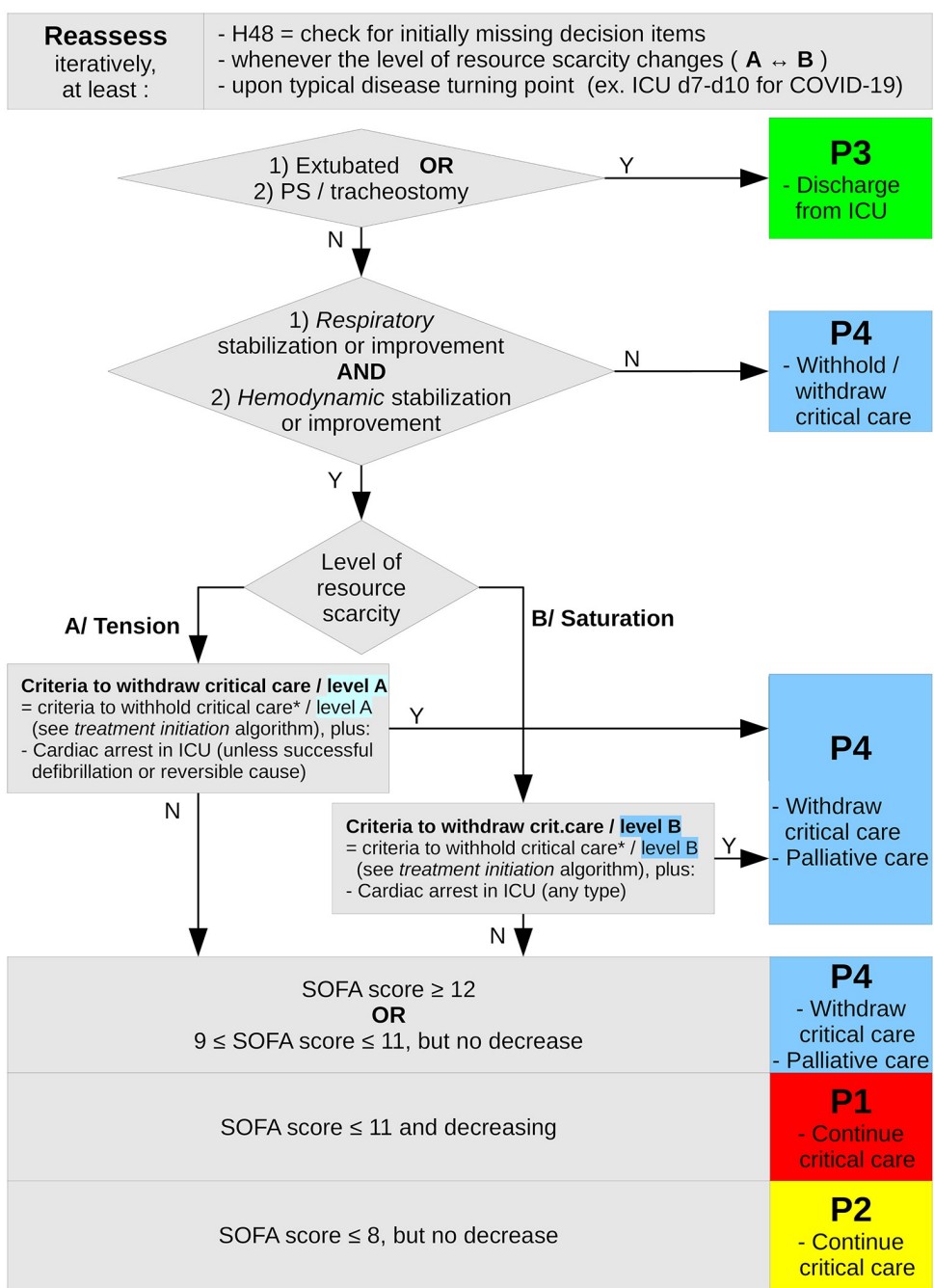

**Fig 2. Triage algorithm for critical care continuation under resource scarcity due to COVID-19.** Summary of the second step (day 7–10 or typical disease turning point, critical care continuation) of the SFAR/SSA critical care prioritization/triage protocol, adapted from [17]. * **Note**: Initial criteria to withhold critical care may have been unknown due to missing information. They should be reassessed in view of the **updated** level of resource scarity.

delivered with ventilators. The status at the end of follow-up (still in ICU, dead or discharged alive) with corresponding date was finally recorded.

### Retrospective priority allocation

According to the SFAR/SSA triage protocol, all patients were retrospectively categorized in 4 levels of priority (P1 to P4) in each situation (tension or saturation), at ICU admission then again after 7 to 10 days for those still in ICU at that time, using a computerized implementation of the protocol (R scripts available on request). To that purpose, in agreement with clinical practice in a triage situation with partly available information, missing values of corresponding variables were imputed as most favorable values, namely absence of corresponding comorbidity or organ dysfunction.

### Primary and secondary outcomes

The pre-specified primary outcome was raw mortality compared between P4 patients and all other priority levels taken together, based on triage simulated on day 0. Secondary outcomes included mortality compared between P4 patients and all other priority levels taken together based on simulated re-triage on day 7 to 10, mortality compared between all priority levels at both time points, cumulative incidence of death and of alive discharge from ICU over time, length of ICU stay and length of ventilation.

Lives potentially lost and saved under triage were also quantified in a simulated situation of saturation. Briefly, under the conservative assumption that all P4 patients would ultimately die without critical care, two-step (day 0, then day 7 to 10) triage decisions were retrospectively applied to the cohort, and outcome and length of stay were corrected accordingly. Besides the corresponding number of potentially avoidable supplementary deaths, the resulting resources made available (number of ICU patient days) were then used to calculate how many supplementary P1 and P2 patients could then have been admitted and, based on their survival recalculated under this two-step triage, how many of them would have ultimately survived. Detailed description of this process is provided in supporting information (S1 Appendix).

### Statistical analyses

All statistical analyses were performed with R statistical software version 3.6.3 (https://cran.r-project.org).

Categorical data are summarized as absolute number and percentage, with 95% confidence interval (95CI) when appropriate. Numerical data are summarized as mean ± standard deviation if normally distributed, checked with Shapiro-Wilk normality test, else as median with inter-quartile range (IQR, first and third quartiles).

For categorical variables, proportions were compared between groups such as priority levels using Pearson's chi-squared or Fisher's exact test as appropriate. Numerical variables were compared between groups using analysis of variance (ANOVA) and Fisher's F test (Student's t test when only two groups) for normally distributed data, otherwise with Kruskal-Wallis non parametric test (Wilcoxon-Mann-Whitney test when only two groups).

For survival analysis, Kaplan-Meier method was not applicable because alive discharge violates the non informative censoring assumption. Death and alive discharge were therefore handled as mutually exclusive competing risks, and their probability of occurrence was estimated with cumulative incidence functions in a competing risk analysis, using cmprsk R package (https://cran.r-project.org/package=cmprsk) [21]. Probabilities of ICU survival, defined as 1 – probability of death in ICU, and of alive discharge from ICU were compared using Gray's test [22].

All tests were two-sided. Differences were considered significant for p-values 0.05 or below.

## Sample size

Because the study was designed before the end of the first wave of COVID-19 outbreak in France, the sample size determination was based on partly arbitrary estimates. With 95% confidence and 80% power, assuming an overall mortality of 30% and a proportion of 15% patients retrospectively triaged as P4, 199 patients were needed to show an association between P4 and a mortality rate twice as high as for all other priority levels taken together. Based on an estimated 70 ICU patients in each center, the study was feasible in three centers with an overall supplementary 5% safety margin.

## Results

### Patients

225 patients were included, respectively 69, 76 and 80 patients from centers 1, 2 and 3. The first patient was admitted to ICU in center 1 on March 6[th], 2020, and the last one was admitted to ICU in center 3 on April 24[th], 2020.

Baseline demographic data and their comparison between survivors (discharged or still in ICU) and non-survivors are shown in Table 1. Patients were predominantly male, overweight or obese, and with known arterial hypertension. One third of them had type 2 diabetes. None of these features differed between survivors and non-survivors. Non-survivors were older and had a higher rate of COPD, higher clinical frailty scores, and higher SAPS2.

151 patients were still in ICU and hence reassessed between day 7 and 10, with an actual median reassessment time of 10 days (IQR 9–10 days). At the end of the follow-up (June 22[nd], 2020), with a maximum follow-up time of 72 days, 61 patients had died, 26 were still in ICU, and 138 had been discharged alive from ICU. This corresponded to an overall 27% censored and 31% uncensored ICU mortality rate.

### Simulated triage

At ICU admission under saturation, out of a total of 225 patients, 149 patients would have been categorized P1, 29 patients P2, 17 patients P3, and 30 patients P4. Among these 30 P4 patients, 25 would have been so triaged based on clinical condition alone, two based on a SOFA score above 12 alone, and three based on both. After reassessment around day 10, out of 151 patients still in ICU at that time, 58 patients would have been categorized P1, 13 patients P2, 34 patients P3, and 46 patients P4. Tables 2 and 3 summarize patient characteristics, severity status and outcome by priority level, respectively on admission and at reassessment around day 10 under saturation. Distribution of day 0 priority levels differed between reassessment priority levels around day 10 (Table 3, chi-squared p = 0.022), without identifiable pattern of association.

Only results for situations of saturation are detailed here. All results for situations of tension are available in supporting information (S1 and S2 Tables).

### Mortality

The pre-specified primary outcome, mortality rate compared between P4 and other priority levels assigned on ICU admission under saturation, was 53% among P4 patients (95CI 34–72%) versus 23% among P1 to P3 patients taken together (95CI 17–30%, chi-squared p = 5.2e-4).

**Table 1. Baseline characteristics and comparison of survivors and non-survivors.**

|  | Survivors (N = 164) | Non-survivors (N = 61) | Total (N = 225) | p value |
|---|---|---|---|---|
| **Age (years)** | 60.9 (11.9) | 68.9 (9.9) | 63.1 (11.9) | < 0.001[1] |
| **Gender** |  |  |  | 0.609[2] |
| F | 45 (27%) | 14 (23%) | 59 (26%) |  |
| M | 119 (73%) | 47 (77%) | 166 (74%) |  |
| **BMI** |  |  |  | 0.176[1] |
| N missing | 0 | 3 | 3 |  |
| Mean (SD) | 29.2 (6.6) | 30.6 (6.7) | 29.6 (6.6) |  |
| **Hypertension** | 88 (54%) | 37 (61%) | 125 (56%) | 0.369[2] |
| **Type 2 diabetes** | 47 (29%) | 19 (31%) | 66 (29%) | 0.743[2] |
| **COPD** |  |  |  | 0.033[2] |
| No | 154 (94%) | 51 (84%) | 205 (91%) |  |
| Gold 1–3 / A-C | 8 (5%) | 8 (13%) | 16 (7%) |  |
| Gold 4 / D | 2 (1%) | 2 (3%) | 4 (2%) |  |
| **Chronic kidney failure** |  |  |  | 1.000[2] |
| No | 155 (95%) | 58 (95%) | 213 (95%) |  |
| KDIGO G3-4 | 6 (4%) | 2 (3%) | 8 (4%) |  |
| KDIGO G5 | 3 (2%) | 1 (2%) | 4 (2%) |  |
| **Severe neurological impairment** | 1 (1%) | 1 (2%) | 2 (1%) | 0.470[2] |
| **Liver cirrhosis** |  |  |  |  |
| No | 163 (99%) | 61 (100%) | 224 (100%) |  |
| A1-5 | 1 (1%) | 0 (0%) | 1 (0%) |  |
| A6 or B7 | 0 (0%) | 0 (0%) | 0 (0%) |  |
| B8-9 or C | 0 (0%) | 0 (0%) | 0 (0%) |  |
| **Clincal frailty score** |  |  |  | 0.005[2] |
| <5 | 158 (96%) | 52 (85%) | 210 (93%) |  |
| 5–6 | 4 (2%) | 8 (13%) | 12 (5%) |  |
| >7 | 2 (1%) | 1 (2%) | 3 (1%) |  |
| **Onset to ICU time (days)** |  |  |  | 0.166[3] |
| N missing | 3 | 3 | 6 |  |
| Median [Q1—Q3] | 8 [5–11] | 7 [4–10] | 7 [5–11] |  |
| **SAPS2** | 36 [27–46] | 47 [40–57] | 40 [29–49] | < 0.001[3] |

Baseline demographic features of the whole cohort (total), survivors (already discharged alive from ICU, or still in ICU at the end of follow-up) and non-survivors.

[1]. Linear Model ANOVA

[2]. Fisher's Exact Test for Count Data

[3]. Kruskal-Wallis rank sum test

Regarding secondary outcomes, at reassessment around day 10, the mortality rate was 50% among P4 patients (95CI 35–65%) versus 15% among others taken together (95CI 9–24%, chi-squared p = 7.1e-6). When comparing all four priority levels, mortality rates consistently increased in the order P3, P1, P2 and P4 both on admission (Table 2, Fisher's exact test p = 1.6e-5) and at reassessment (Table 3, Fisher's exact test p = 5.6e-5).

## Cumulative incidence analyses

Based on triage simulated on admission and at reassessment around day 10, the cumulative incidence of death was consistently higher (Gray's test, admission p = 3.4e-4, reassessment p<1e-5) and the cumulative incidence of alive ICU discharge consistently lower (admission

**Table 2. Patient severity and outcome by priority levels (day 0, saturation).**

| | P1 (N = 149) | P2 (N = 29) | P3 (N = 17) | P4 (N = 30) | Total (N = 225) | p value |
|---|---|---|---|---|---|---|
| **Age (years)** | 61.8 (11.5) | 64.3 (8.9) | 59.9 (12.6) | 70.0 (13.7) | 63.1 (11.9) | 0.003[1] |
| **Gender** | | | | | | 0.280[2] |
| F | 37 (25%) | 5 (17%) | 6 (35%) | 11 (37%) | 59 (26%) | |
| M | 112 (75%) | 24 (83%) | 11 (65%) | 19 (63%) | 166 (74%) | |
| **BMI** | | | | | | 0.822[1] |
| N missing | 2 | 1 | 0 | 0 | 3 | |
| Mean (SD) | 29.7 (6.7) | 29.6 (4.6) | 28.2 (6.1) | 30.0 (8.3) | 29.6 (6.6) | |
| **Hypertension** | 85 (57%) | 18 (62%) | 3 (18%) | 19 (63%) | 125 (56%) | 0.010[2] |
| **Type 2 diabetes** | 45 (30%) | 7 (24%) | 2 (12%) | 12 (40%) | 66 (29%) | 0.216[2] |
| **SAPS2** | 37 [27–48] | 47 [37–62] | 24 [18–31] | 47 [36–62] | 40 [29–49] | < 0.001[3] |
| **Onset to ICU time (days)** | | | | | | 0.024[3] |
| N missing | 4 | 2 | 0 | 0 | 6 | |
| Median [Q1—Q3] | 8 [6–11] | 7 [5–10] | 7 [5–12] | 5 [3–8] | 7 [5–11] | |
| **SOFA score (day 0)** | 5 [3–6] | 9 [8–10] | 2 [2–3] | 7 [4–9] | 5 [3–7] | |
| **Mechanical ventilation** | 123 (83%) | 29 (100%) | 0 (0%) | 24 (80%) | 176 (78%) | |
| **PaO2/FiO2** | | | | | | |
| N missing | 2 | 0 | 0 | 0 | 2 | |
| Median [Q1—Q3] | 125 [93–160] | 117 [83–157] | 272 [222–300] | 118 [96–142] | 125 [94–178] | |
| **Vasopressors** | | | | | | |
| No | 106 (71%) | 1 (3%) | 17 (100%) | 15 (50%) | 139 (62%) | |
| Dobutamine only | 1 (1%) | 0 (0%) | 0 (0%) | 0 (0%) | 1 (0%) | |
| Epi-/Norepinephrine < 0.1 μg/kg/min | 18 (12%) | 3 (10%) | 0 (0%) | 4 (13%) | 25 (11%) | |
| Epi-/Norepinephrine > 0.1 μg/kg/min | 24 (16%) | 25 (86%) | 0 (0%) | 11 (37%) | 60 (27%) | |
| **Glasgow coma score** | | | | | | |
| N missing | 1 | 0 | 0 | 0 | 1 | |
| Median [Q1—Q3] | 15 [15–15] | 15 [15–15] | 15 [15–15] | 15 [15–15] | 15 [15–15] | |
| **Serum creatinine (μmol/L)** | | | | | | |
| N missing | 1 | 0 | 0 | 0 | 1 | |
| Median [Q1—Q3] | 72 [60–89] | 110 [83–146] | 76 [64–93] | 112 [59–228] | 77 [61–100] | |
| **Platelet count (10^9/L)** | | | | | | |
| N missing | 1 | 0 | 0 | 0 | 1 | |
| Median [Q1—Q3] | 224 [183–292] | 156 [124–263] | 250 [190–293] | 176 [136–255] | 215 [167–289] | |
| **Bilirubin (μmol/L)** | | | | | | |
| N missing | 8 | 1 | 1 | 1 | 11 | |
| Median [Q1—Q3] | 10.0 [7.0–14.0] | 10.5 [8.0–14.0] | 8.5 [6.0–12.0] | 9.0 [7.0–14.0] | 10.0 [7.0–13.8] | |
| **ECMO (day 0)** | | | | | | |
| No | 149 (100%) | 29 (100%) | 17 (100%) | 29 (97%) | 224 (100%) | |
| VV | 0 (0%) | 0 (0%) | 0 (0%) | 1 (3%) | 1 (0%) | |
| VA | 0 (0%) | 0 (0%) | 0 (0%) | 0 (0%) | 0 (0%) | |
| **Mortality** | 32 (21%) | 13 (45%) | 0 (0%) | 16 (53%) | 61 (27%) | < 0.001[2] |
| **Discharged alive** | 99 (66%) | 9 (31%) | 17 (100%) | 13 (43%) | 138 (61%) | < 0.001[2] |
| **Raw length of ICU stay (days)** | 11 [6–21] | 16 [7–30] | 3 [2–12] | 6 [4–16] | 11 [5–21] | 0.001[3] |
| **Length of ventilation (days)** | 10 [4–17] | 16 [6–26] | 0 [0–7] | 6 [2–14] | 9 [4–17] | < 0.001[3] |

*(Continued)*

**Table 2.** (Continued)

| | P1 (N = 149) | P2 (N = 29) | P3 (N = 17) | P4 (N = 30) | Total (N = 225) | p value |
|---|---|---|---|---|---|---|
| **Treatment withheld or withdrawn** | 20 (13%) | 6 (21%) | 0 (0%) | 14 (47%) | 40 (18%) | < 0.001[2] |

Patient severity and outcome according to the priority level assigned at ICU admission (day 0, first step of the SFAR/SSA critical care triage protocol) in a situation of saturation of critical care capacities.

[1]. Linear Model ANOVA

[2]. Fisher's Exact Test for Count Data

[3]. Kruskal-Wallis rank sum test

p = 0.017, reassessment p = 3.4e-5) in P4 than in other priority levels taken together (Fig 3). Accordingly, the cumulative incidence of death consistently increased in the order P3, P1, P2 and P4 between all four priority levels set on admission (Gray's test, admission p = 3.1e-5) and at reassessment (p = 8e-5). The cumulative incidence of alive ICU discharge also differed between all priority levels at both time points with a consistent increase in the order P4, P1 and P3 but variations in time for P2 (both p<1e-5) (Fig 4).

### Resource utilization

The raw length of ICU stay and length of ventilation by priority level on admission are summarized in Tables 2 and 3 at corresponding time points. All significantly differed between priority levels (Kruskal-Wallis tests, p≤0.001) with various difference patterns. The lengths of ICU stay and ventilation were shortest for P3 patients both on day 0 and at reassessment around day 10, and longest for P2 patients on day 0 (Figs 5 and 6).

### Quantification of lives potentially saved

Under lasting saturation of critical care capacity, following the SFAR/SSA triage protocol both initially and at reassessment around day 10 would have led to withhold or withdraw critical care in P4 patients, hence 32 supplementary avoidable deaths among the 225 study patients. Conversely, this would have made resources available (1225 ICU patient days) to treat 92 more P1 or P2 patients with a very high probability of death without critical care, 60 of whom would have ultimately survived under two-step triage, resulting in an overall estimated 28 more lives potentially saved than without triage. In the hypothesis of an even worse overwhelming situation, having to withhold or withdraw critical care in all but P1 patients would have similarly allowed an estimated 40 more lives potentially saved (52 supplementary avoidable deaths, 1732 ICU days made available, 144 more P1 patients treated, 92 of whom survivors).

### Effect of SAPS2, age and center

The distribution of SAPS2 and age by priority levels, their association with outcome and their differences between centers were studied as robustness analyses (Figs 7–11). When compared between priority levels at admission, SAPS2 was lowest in P3, intermediate in P1 and highest in P2-P4 patients (Kruskal-Wallis, p = 1.1e-7, Fig 7). The cumulative incidence of death consistently increased and that of alive ICU discharge consistently decreased with SAPS2 quartiles (Gray's test, both p<1e-5), with minimal differences below the median SAPS2 of 40 (Fig 8). The age distribution between priority levels is shown in Fig 9. The cumulative incidence of death increased (p = 0.0016) and that of alive ICU discharge decreased (p = 7.8e-5) with age quartiles (Fig 10). The age of patients was similar among centers (Kruskal-Wallis, p = 0.9) but the SAPS2 distribution differed as lower SAPS2 were observed in center 3 (P = 4.1e-4).

**Table 3. Patient severity and outcome by priority levels (day 7–10, saturation).**

| | P1 (N = 58) | P2 (N = 13) | P3 (N = 34) | P4 (N = 46) | Total (N = 151) | p value |
|---|---|---|---|---|---|---|
| **Age (years)** | 64.0 (8.4) | 63.4 (11.9) | 62.5 (13.0) | 63.1 (11.8) | 63.3 (10.9) | 0.941[1] |
| **Gender** | | | | | | 0.703[2] |
| F | 18 (31%) | 2 (15%) | 8 (24%) | 12 (26%) | 40 (26%) | |
| M | 40 (69%) | 11 (85%) | 26 (76%) | 34 (74%) | 111 (74%) | |
| **BMI** | | | | | | 0.058[1] |
| N missing | 0 | 1 | 0 | 0 | 1 | |
| Mean (SD) | 30.2 (5.4) | 32.3 (11.6) | 27.6 (5.4) | 31.4 (7.4) | 30.2 (6.8) | |
| **Hypertension** | 38 (66%) | 9 (69%) | 20 (59%) | 22 (48%) | 89 (59%) | 0.281[2] |
| **Type 2 diabetes** | 19 (33%) | 5 (38%) | 9 (26%) | 17 (37%) | 50 (33%) | 0.750[2] |
| **SAPS2** | 46 [36–53] | 31 [27–39] | 36 [29–48] | 41 [30–54] | 41 [31–50] | 0.021[3] |
| **Onset to ICU time (days)** | | | | | | 0.400[3] |
| N missing | 3 | 1 | 0 | 1 | 5 | |
| Median [Q1—Q3] | 7 [6–10] | 8 [3–10] | 9 [5–11] | 7 [5–9] | 7 [5–10] | |
| **Priority level, day 0 (saturation)** | | | | | | 0.022[4] |
| P1 | 37 (64%) | 10 (77%) | 28 (82%) | 34 (74%) | 109 (72%) | |
| P2 | 15 (26%) | 1 (8%) | 1 (3%) | 5 (11%) | 22 (15%) | |
| P3 | 0 (0%) | 1 (8%) | 0 (0%) | 4 (9%) | 5 (3%) | |
| P4 | 6 (10%) | 1 (8%) | 5 (15%) | 3 (7%) | 15 (10%) | |
| **SOFA score (day 7–10)** | 4 [3–5] | 5 [4–6] | 2 [2–3] | 7 [5–9] | 4 [3–6] | |
| **Mechanical ventilation** | 58 (100%) | 13 (100%) | 0 (0%) | 46 (100%) | 117 (77%) | |
| **PaO2/FiO2** | | | | | | |
| N missing | 0 | 0 | 2 | 0 | 2 | |
| Median [Q1—Q3] | 140 [108–180] | 111 [95–215] | 222 [175–270] | 98 [75–130] | 135 [95–196] | |
| **Vasopressors** | | | | | | |
| N missing | 0 | 1 | 0 | 0 | 1 | |
| No | 53 (91%) | 9 (75%) | 31 (91%) | 20 (43%) | 113 (75%) | |
| Dobutamine only | 0 (0%) | 0 (0%) | 0 (0%) | 0 (0%) | 0 (0%) | |
| Epi-/Norepinephrine < 0.1 µg/kg/min | 2 (3%) | 1 (8%) | 1 (3%) | 11 (24%) | 15 (10%) | |
| Epi-/Norepinephrine > 0.1 µg/kg/min | 3 (5%) | 2 (17%) | 2 (6%) | 15 (33%) | 22 (15%) | |
| **Glasgow coma score** | | | | | | |
| N missing | 1 | 0 | 0 | 1 | 2 | |
| Median [Q1—Q3] | 15 [15–15] | 15 [15–15] | 15 [15–15] | 15 [15–15] | 15 [15–15] | |
| **Serum creatinine (µmol/L)** | 69 [52–95] | 72 [59–119] | 64 [54–79] | 99 [61–181] | 72 [56–109] | |
| **Platelet count (10^9/L)** | 360 [280–427] | 358 [279–418] | 400 [272–456] | 322 [221–416] | 355 [252–447] | |
| **Bilirubin (µmol/L)** | | | | | | |
| N missing | 14 | 6 | 7 | 9 | 36 | |
| Median [Q1—Q3] | 8.0 [6.0–11.0] | 10.0 [6.0–26.1] | 8.0 [6.0–11.0] | 9.0 [6.8–13.0] | 8.0 [6.0–12.0] | |
| **ECMO (day 7–10)** | | | | | | |
| No | 49 (84%) | 12 (92%) | 33 (97%) | 44 (96%) | 138 (91%) | |
| VV | 9 (16%) | 1 (8%) | 1 (3%) | 2 (4%) | 13 (9%) | |
| VA | 0 (0%) | 0 (0%) | 0 (0%) | 0 (0%) | 0 (0%) | |
| **Mortality** | 10 (17%) | 4 (31%) | 2 (6%) | 23 (50%) | 39 (26%) | < 0.001[2] |
| **Discharged alive** | 35 (60%) | 7 (54%) | 31 (91%) | 18 (39%) | 91 (60%) | < 0.001[2] |
| **Raw length of ICU stay (days)** | 20 [13–37] | 13 [12–22] | 10 [8–13] | 19 [13–29] | 16 [11–28] | < 0.001[3] |
| **LOS after reassessment (days)** | 10 [3–27] | 4 [2–12] | 0 [0–3] | 10 [3–20] | 6 [1–19] | < 0.001[3] |
| **Length of ventilation (days)** | 16 [11–30] | 12 [11–22] | 8 [6–9] | 16 [11–28] | 14 [9–24] | < 0.001[3] |

*(Continued)*

**Table 3.** (Continued)

|  | P1 (N = 58) | P2 (N = 13) | P3 (N = 34) | P4 (N = 46) | Total (N = 151) | p value |
|---|---|---|---|---|---|---|
| **Treatment withheld or withdrawn** | 8 (14%) | 3 (23%) | 2 (6%) | 14 (30%) | 27 (18%) | 0.023[2] |

Patient severity and outcome according to the priority level assigned at reassessment (day 7 to 10, second step of the SFAR/SSA critical care triage protocol) in a situation of saturation of critical care capacities.

1. Linear Model ANOVA

2. Fisher's Exact Test for Count Data

3. Kruskal-Wallis rank sum test

4. Pearson's Chi-squared test

Corresponding differences were observed in cumulative incidence of death and ICU discharge (Fig 11).

## Situations of tension

In a putative situation of tension, the tabulated summarized data of retrospective initial priority allocation, namely patient characteristics, severity status and outcome by priority level, along with comparison of priority allocations between both steps, are given in supporting information (S1 and S2 Tables). Since no recovered cardiac arrest was recorded during initial ICU stay in the study cohort, the second step of priority allocation (reassessment on day 7 to 10) yielded identical priority levels to that obtained for situations of saturation. All corresponding analyses regarding cumulative incidence of death and alive ICU discharge, and

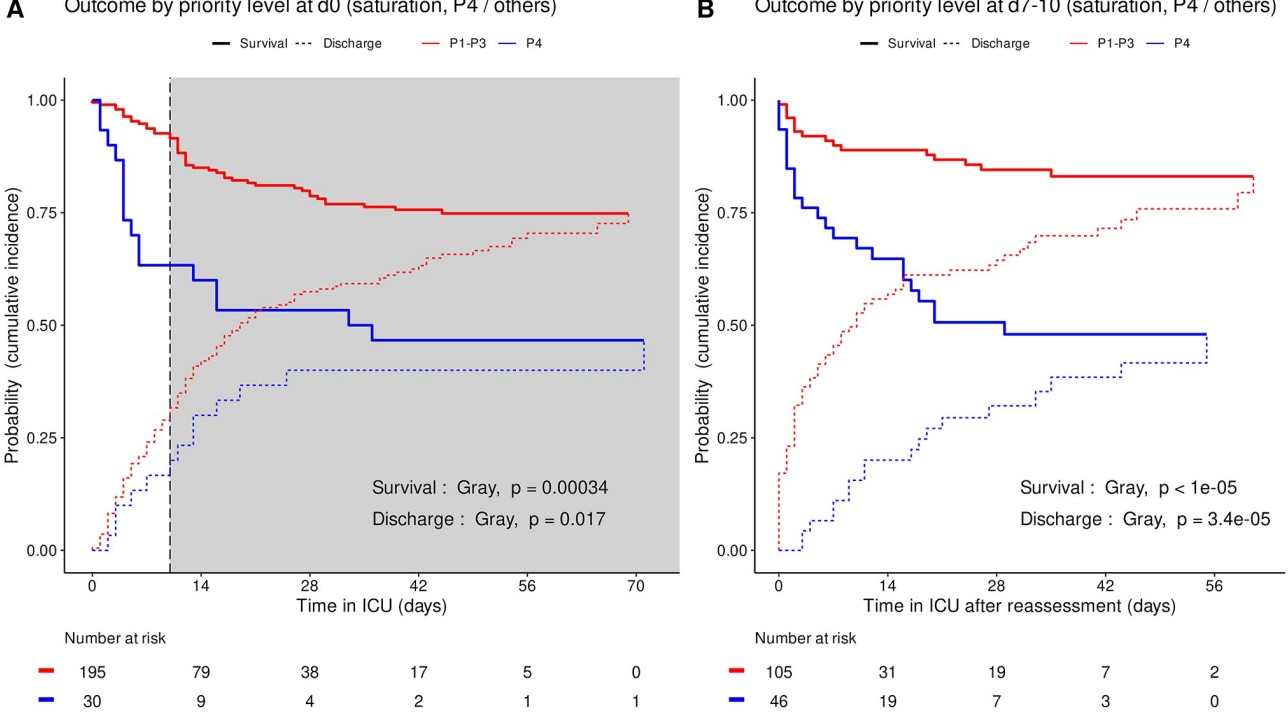

**Fig 3. Outcome of COVID-19 ICU patients by priority level in saturation: P4 *vs.* others.** Cumulative incidence (c.i.) of alive discharge from ICU and survival (= 1 –c.i. of death in ICU) for COVID-19 patients: P4 compared with other priority levels at day 0 (A), and at reassessment on day 7 to 10 (B). Shaded area: initial prioritization no longer relevant due to reassessment.

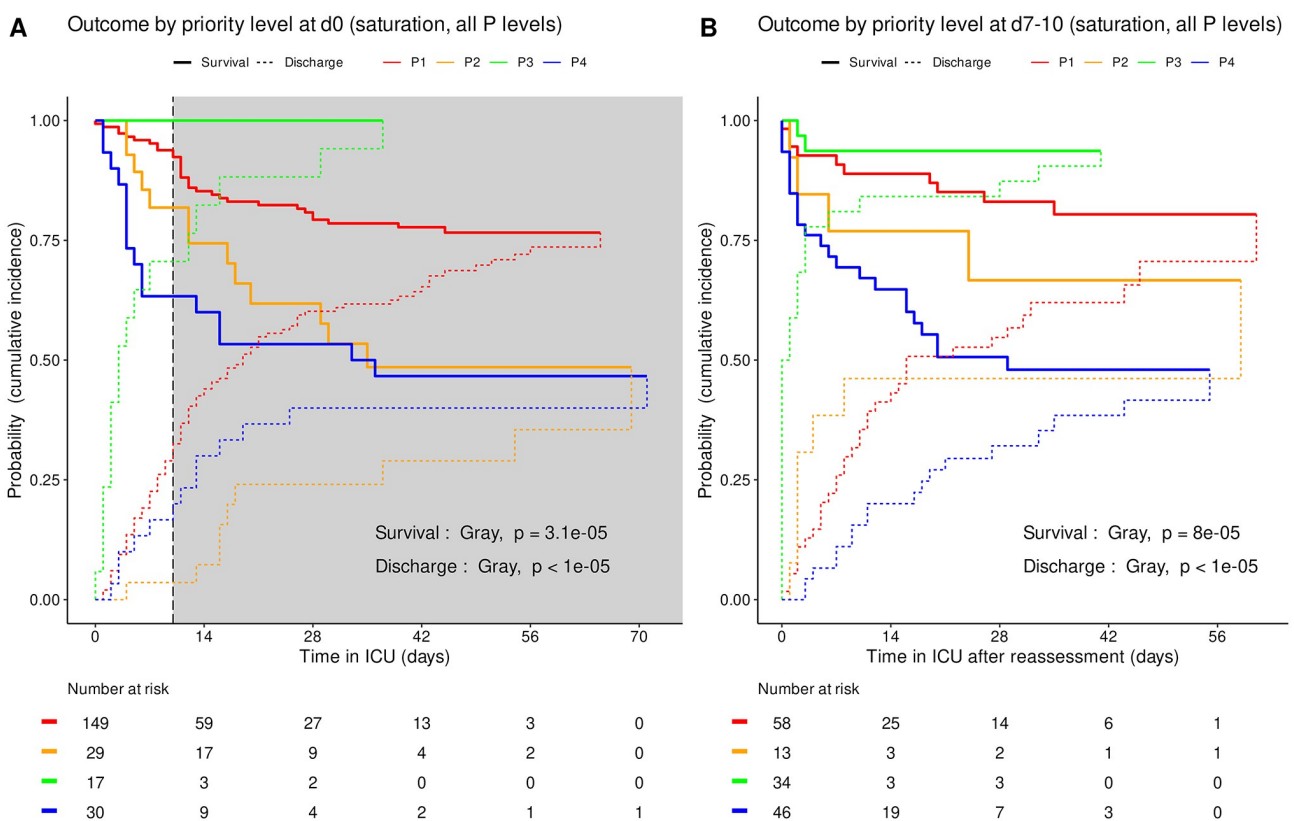

**Fig 4. Outcome of COVID-19 ICU patients by priority level in saturation: All priority levels.** Cumulative incidence (c.i.) of alive discharge from ICU and survival (= 1 −c.i. of death in ICU) for COVID-19 patients: comparison between all priority levels at day 0 (A) and at reassessment on day 7 to 10 (B). Shaded area: initial prioritization no longer relevant due to reassessment.

resource utilization in terms of length of ICU stay and length of ventilation are given in supporting information (S1–S3 Figs).

## Discussion

In this multi-center cohort study, using the SFAR/SSA critical care triage protocol for situations of saturation, retrospective simulated triage of 225 patients admitted to ICUs for severe COVID-19 appropriately classified them by probability of survival.

### Validation of the triage protocol

Beyond this clinical validation of the classification ability of the SFAR/SSA triage protocol, our study documents its ability to actually help save more lives if critical care capacities are saturated or overwhelmed.

Our results support the design of the SFAR/SSA algorithm, by which triage mainly relies on pre-existing clinical condition initially, then on SOFA score, its components and its evolution in time, as inspired from a former triage algorithm designed for an influenza pandemic [15, 24]. Its two phase design with initial prioritization revised after 7 to 10 days as the typical disease turning point appeared relevant. In our cohort indeed, early differences in probability of survival between priority levels appeared less consistent after day 10, possibly due to the smaller number of patients still in ICU and to the impact of intercurrent events on outcome (Fig 4A). Day 10 reassessment consistently predicted later outcome (Fig 4B).

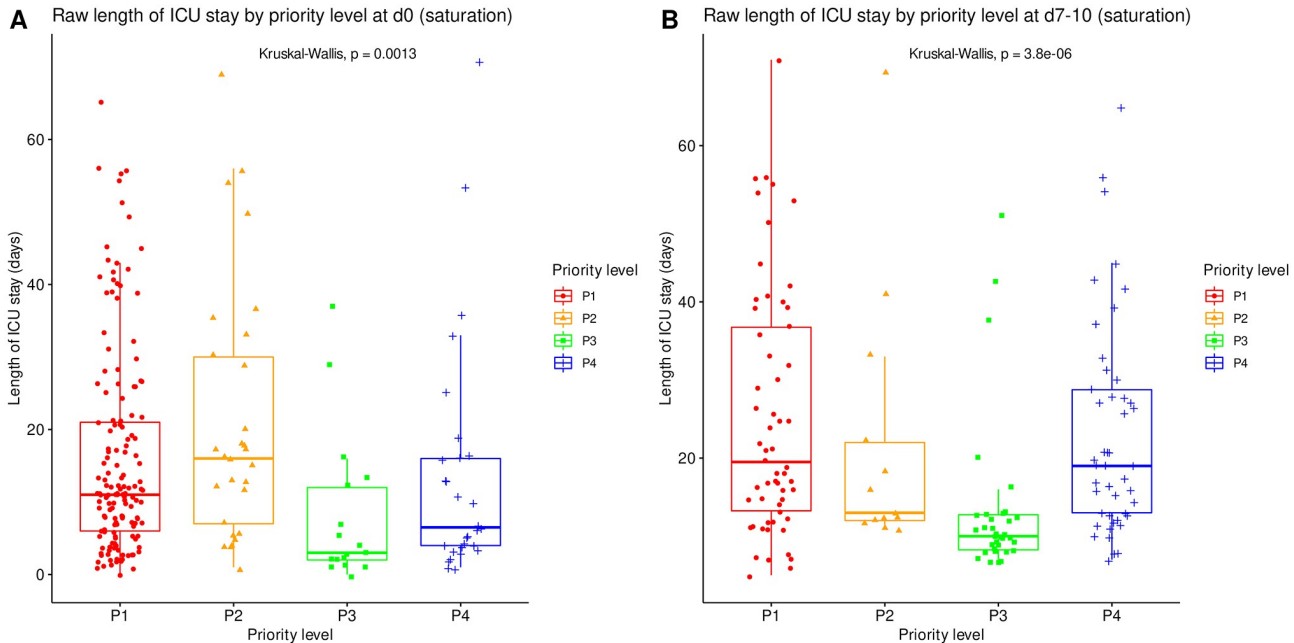

**Fig 5. Raw length of ICU stay by priority level in saturation.** Length of ICU stay, irrespective of patient outcome, compared between all priority levels at day 0 (A, N = 225) and at reassessment on day 7 to 10 (B, N = 151). Boxes: median, 1st and 3rd quartiles; whiskers: Tukey's convention (farthest points within 1.5 x IQR distance from box).

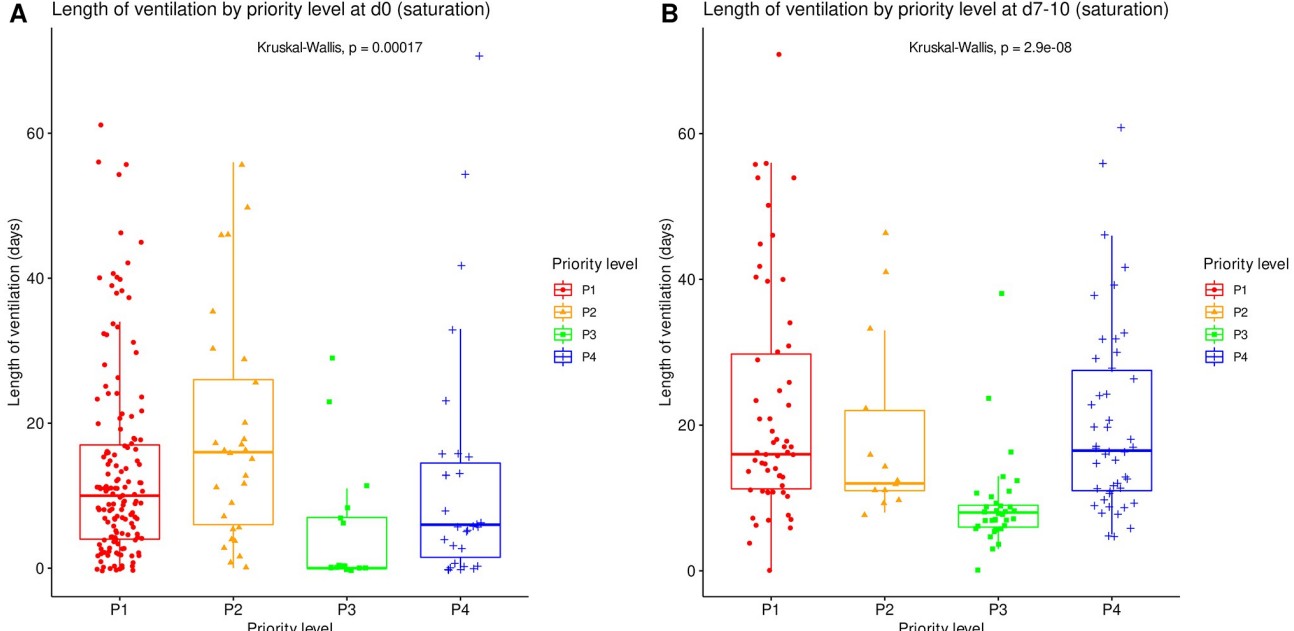

**Fig 6. Length of mechanical ventilation by priority level in saturation.** Length of mechanical ventilation, irrespective of patient outcome, compared between all priority levels at day 0 (A, N = 225) and at reassessment on day 7 to 10 (B, N = 151). Boxes: median, 1st and 3rd quartiles; whiskers: Tukey's convention (farthest points within 1.5 x IQR distance from box).

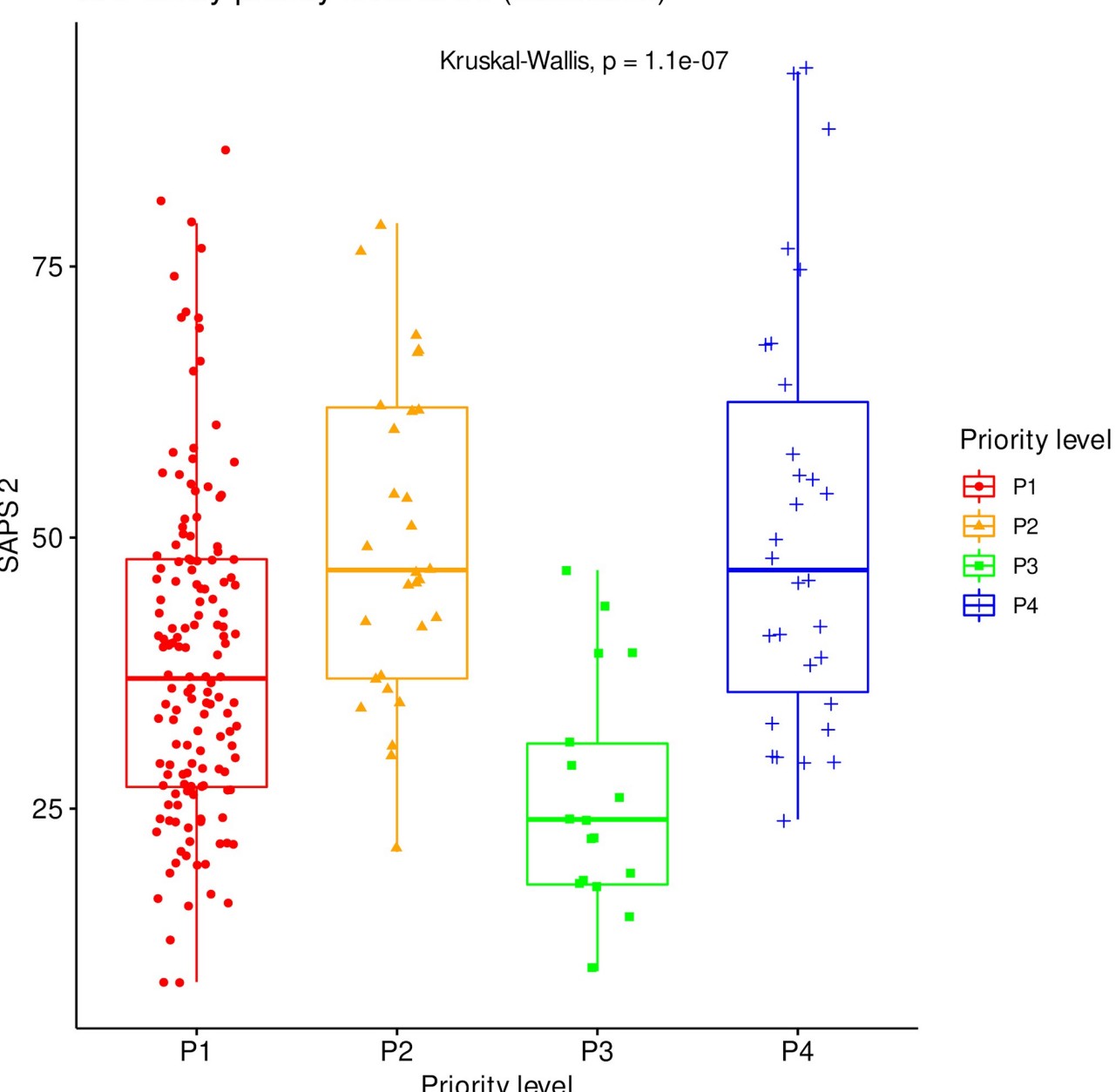

**Fig 7. SAPS2 distribution by initial priority level in saturation.** SAPS2 compared between all priority levels at day 0 (N = 225). Boxes: median, 1$^{st}$ and 3$^{rd}$ quartiles; whiskers: Tukey's convention (farthest points within 1.5 x IQR distance from box).

Patients retrospectively assigned to P4 or P3 were deliberately kept within the cohort for outcome analyses. During such a crisis, while ICU capacity is being extended, no triage applies yet and ICUs can also admit patients who would fulfill P3 or P4 criteria, as observed in our study. If triage later becomes necessary, it applies both to critical care initiation and continuation to avoid "first arrived, first served" unethical discrimination [17]. Patients then undergo the second triage step even though they were not initially triaged at first step. This approach therefore relevantly assesses the classification ability of the triage algorithms.

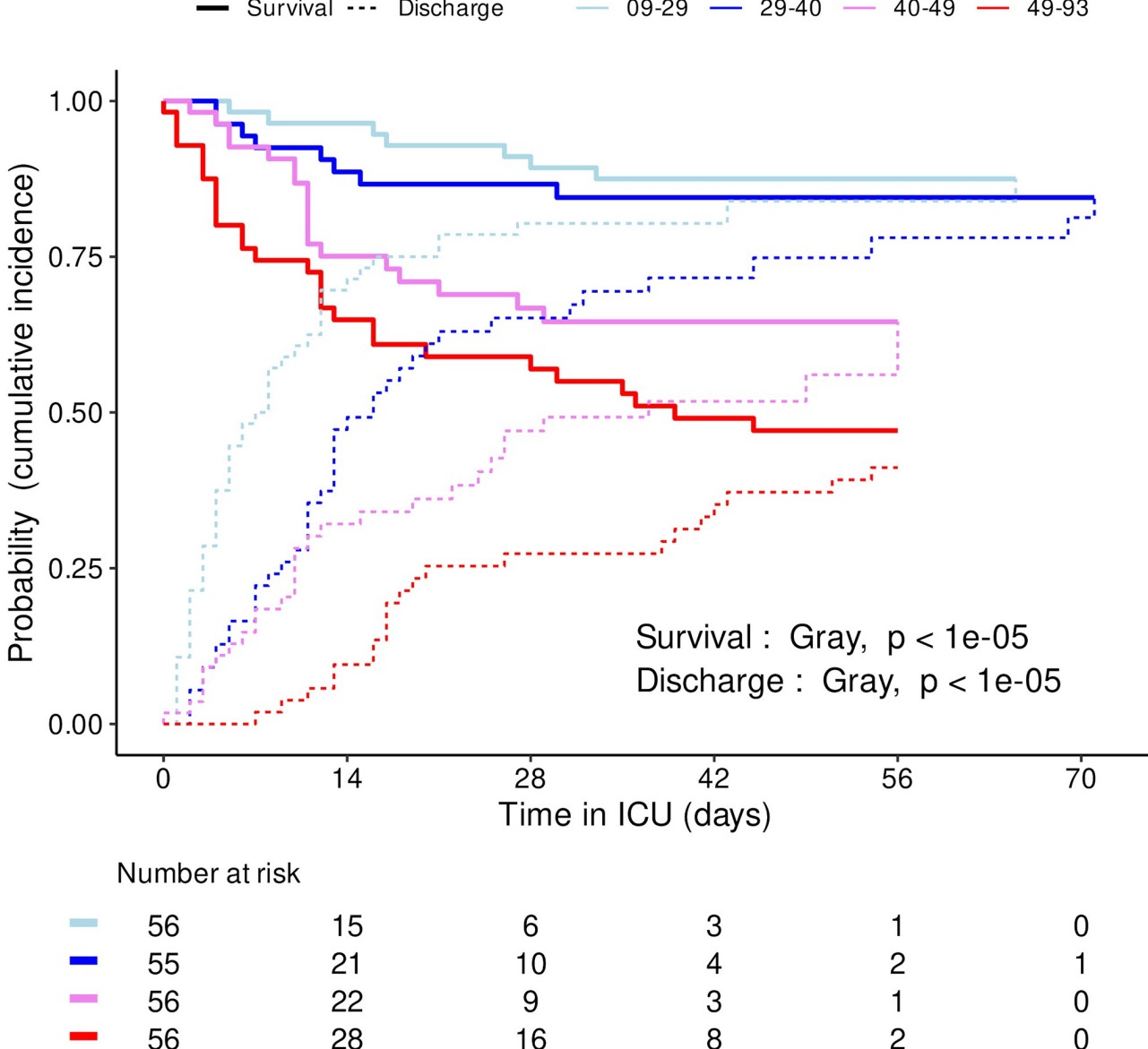

**Fig 8. Outcome of COVID-19 ICU patients by SAPS2 quartile.** Cumulative incidence (c.i.) of alive discharge from ICU and survival (= 1 −c.i. of death in ICU) for COVID-19 patients: comparison between SAPS2 quartiles.

Conversely, to quantify lives potentially saved, the triage protocol was strictly applied to the retrospective cohort with corresponding exclusion of P3 and P4 patients, in order to estimate its performance at full scale including under prolonged implementation due to lasting saturation of critical care capacity.

The absence of association between obesity, arterial hypertension or diabetes and ICU outcome (Table 1) despite their high prevalence and although they are known factors of severe COVID-19 also support the choice not to include them in triage criteria [17, 25]. The prevalence of other pre-existing comorbidities was low (Table 1). Although no large scale

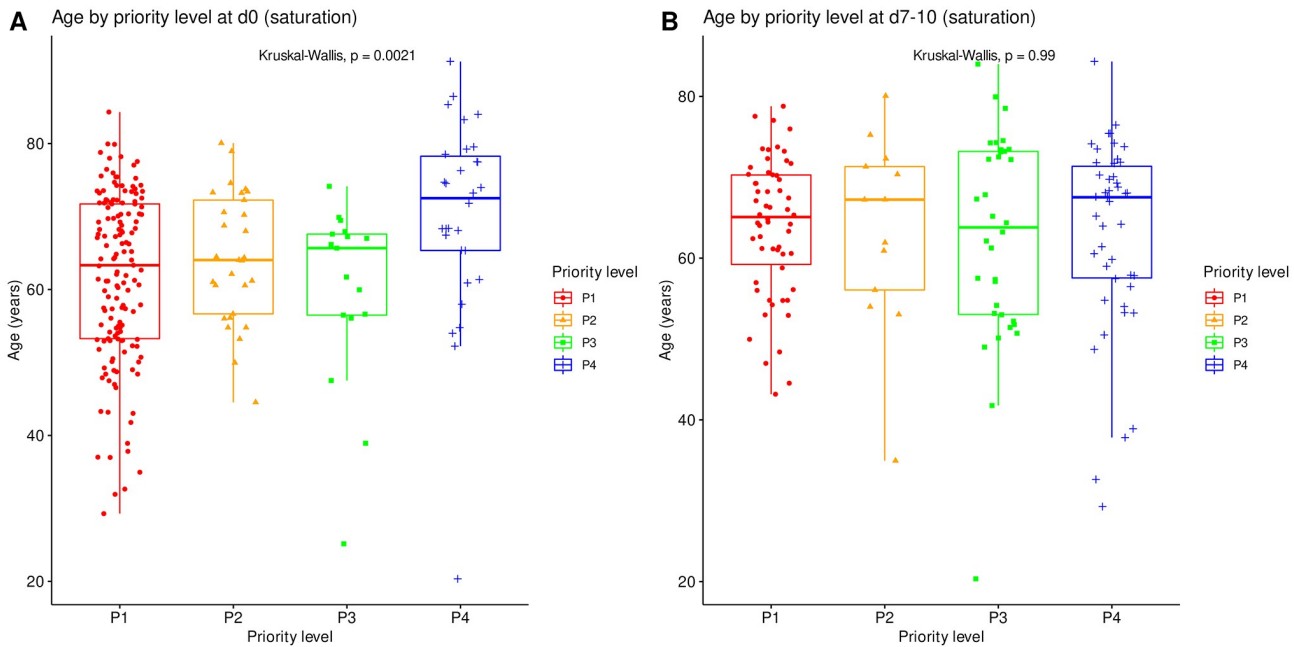

**Fig 9. Age distribution by initial priority level in saturation.** Age compared between all priority levels at day 0 (N = 225). Boxes: median, 1st and 3rd quartiles; whiskers: Tukey's convention (farthest points within 1.5 x IQR distance from box).

systematized triage was implemented during the first epidemic wave of COVID-19 in France, this likely reflects the role of these comorbidities in individual critical care withholding or withdrawal decisions, in line with national and international recommendations [26, 27]. Our study had insufficient power to assess their individual relevance to ICU prognosis in severe COVID-19 hence to triage. This point will be studied in a large cohort in an ancillary study of the French COVID-ICU registry [25].

This study was not designed to compare the SFAR/SSA triage protocol with other potential outcome predictors such as age and SAPS2 but their association with outcome was explored. Consistently with SAPS2 design, its highest values were associated with worst outcome in our cohort [28]. Median SAPS2 of 40 was an apparent shift point in mortality, lower values showing limited discriminating ability (Fig 8). Since SAPS2 cannot be calculated before 24h after ICU admission, this obviously precludes its use as a triage tool for critical care initiation. Higher age was also associated with mortality (Table 1), in line with epidemiological data about COVID-19 [29]. However, cumulative incidence analyses by age quartiles suggested that this association was not fully consistent in time and that there were discrepancies regarding probability of alive discharge by age, possibly reflecting health status heterogeneity within age categories (Fig 10). These results support the choice to include age among triage criteria but not to rely on it alone [17]. To that regard, "age ≥ 85 years" as a P4 criterion among others appears conservative and epidemiologically relevant to save the largest number of lives, yet legally problematic in countries where it could violate anti-discrimination laws. Using "age ≥ 85 years with at least 1 comorbidity" as a substitute might solve this issue without altering the usability and performance of the triage protocol.

Finally, a potential center effect appears unlikely or marginal. Age distribution was similar among centers (Fig 11B), but survival and alive discharge were higher in center 3 than in centers 1–2 (Fig 11C). This could be explained by a difference in patient severity (Fig 11A) but

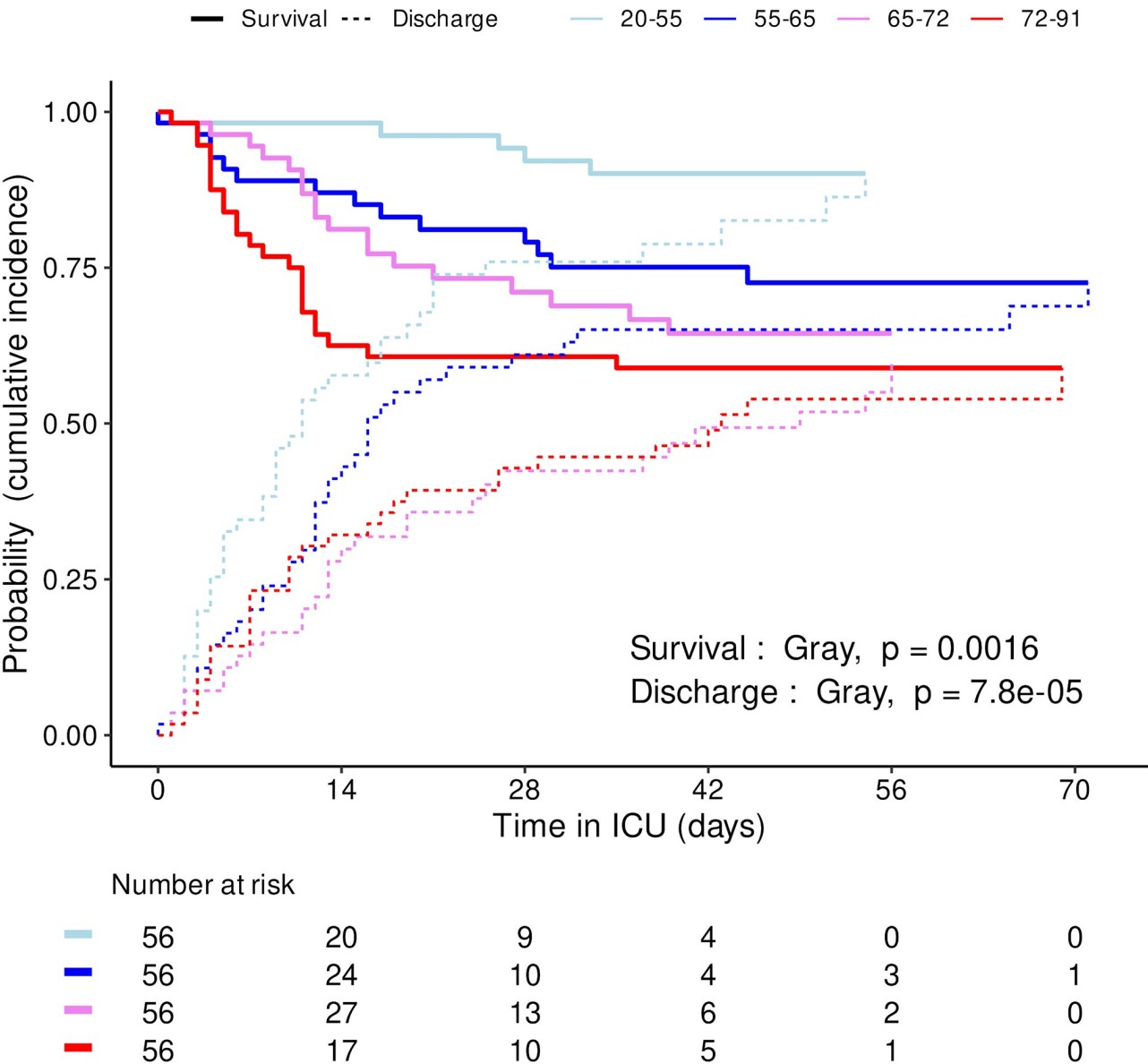

**Fig 10. Outcome of COVID-19 ICU patients by age quartile.** Cumulative incidence (c.i.) of alive discharge from ICU and survival (= 1 –c.i. of death in ICU) for COVID-19 patients: comparison between age quartiles.

also by epidemiological features. The first wave of the COVID-19 outbreak actually followed a westbound spread in France, which may have helped achieve better readiness in the latest affected ICUs as the healthcare system improved its response. This is supported by the observation that, in a large national cohort of 4244 ICU patients with COVID-19 during the first epidemic wave in France, mortality decreased over the study period [25]. A Hawthorne effect is unlikely in this study as SFAR/SSA triage recommendations were published online right at the time when lock-down began to produce its effect with a drop in COVID-19 ICU admissions.

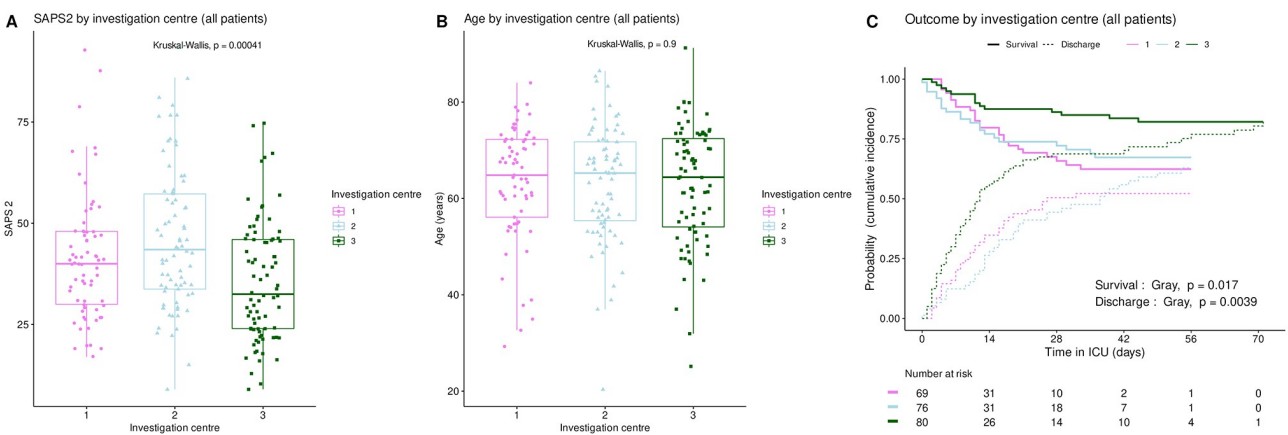

**Fig 11. Initial severity, age and outcome compared between investigation centers.** Inter-center variations in SAPS2 (A), age (B), and ICU outcome (C). Boxplots: box–median, 1$^{st}$ and 3$^{rd}$ quartiles; whiskers–Tukey's convention, farthest points within 1.5 x IQR distance from box. C: cumulative incidence (c.i.) of alive discharge from ICU and survival (= 1 –c.i. of death in ICU).

## Study strengths

First, this study provides evidence that the SFAR/SSA triage protocol is usable and relevant in a COVID-19 outbreak with resource scarcity without significant immunity in the population. To our knowledge, among many proposed triage schemes to prioritize scarce critical care resources for COVID-19 patients with highest probability of benefit in order to maximize the number of saved lives in agreement with national and international ethical guidelines, it is the first one to obtain *a posteriori* clinical validation [13, 16, 30–38].

Second, our study did not only assess the ability of the SFAR/SSA triage protocol to predict outcome, it also quantified the potential number of lives saved. This could help authorities make informed decisions and population understand that triage is not about denying treatment but about saving more lives. This most important yet conceptually difficult notion needs explanation and advocacy, at best out of crisis situations.

Third, we used competing risk analysis by cumulative incidence. It is the proper way to describe ICU survival because alive discharge is a desired outcome but a competing event that violates Kaplan-Meier's assumption of non informative censoring [21]. Some have advocated that only mortality rate applies in ICU, because irrespective of methods used, prolonged survival does not benefit patients who actually die in ICU [39]. In the present study however, time-to-event analyses make sense from a resource utilization point of view. Facing a saturating patient inflow under scarce resources, longer ICU stays are acceptable only if they actually avoid ICU deaths. Earlier alive discharge increases ICU resource availability for other patients.

## Study limitations

First, this study is retrospective and based on a small sample size. However, its prospective validation was not feasible during the first COVID-19 epidemic wave, and it would have been unethical to wait for a next one to prospectively validate it. However, the fact that patient characteristics in our three-center cohort remarkably matched those in the 4244 patient cohort of the national COVID-ICU study in 138 hospitals over the same period, along with the high similarity of observed outcomes (27% censored and 31% uncensored ICU mortality in our cohort, vs. 31% 90 day-mortality in the COVD-ICU study) further support the very good representativity of our cohort [25]. Our sample size also proved appropriate to achieve the main study goal.

Second, an ideal study would have included all COVID-19 patients for whom critical care was considered at some point during the study period, in order to provide optimal estimates of the spread of patients among priority levels in triage conditions, along with their actual outcome. Of note, due to large scale ICU capacity extension and inter-regional patient transfers, the first epidemic wave of COVID-19 in France resulted in a severe hospital and critical care strain but not in full saturation [19]. As a consequence, no systematized triage was used but only individualized treatment withholding or withdrawal decisions were made. Some patients who would have been triaged P3 or P4 in a situation of saturation thus actually received critical care treatment, although only part of them. Only ICU patients were included in this study for feasibility reasons. Many others were potentially missed: patients not referred to ICU considering unfavorable age and medical history, and patients proposed but not admitted for insufficient severity or after ethical discussion. Tracking them with ICU non admission registries and medical ward records would theoretically allow to check agreement with P3 and P4 triage, but those data were not available. The absence of these patients in our cohort likely underestimated the number of P3 and P4 patients, and the mortality in P4 patients whose outcome with palliative care is unrecorded.

Third, our study was neither designed nor powered to challenge the reassessment date or the values of triage thresholds regarding SOFA score, age or severity of pre-existing comorbidities. This could be justified considering the low P2/P1 ratio in the study. It should be done in the aforementioned ancillary COVID-ICU study.

Fourth, we dealt only with COVID-19 patients, whereas triage would also apply to other critically ill or critically injured patients in a situation of ICU resource scarcity. However, during the first epidemic wave in France as in many other countries, COVID-19 patients were the vast majority of those in need of critical care at that time. Biosafety considerations also led to separate COVID-19 and non-COVID-19 ICU sectors. Actual triage may therefore occur separately between COVID-19 and other conditions.

Fifth, we did not address inter-rater variability of priority level assignment.

Finally, the results for situations of tension were similar to those for saturation but less clear-cut, likely due to ICU admissions being already impacted by actual tension (S1 and S2 Tables, S1–S3 Figs).

## Triage algorithms and Machine Learning

Machine Learning (ML) has been proposed as an alternative approach to develop COVID-19 triage systems. ML describes the use of computer-based adaptive models that are able to mimic learning without additional formal programming instructions, by using algorithms and statistical models to analyze and draw inferences from patterns in patients data. Four major benefits of a ML based (or data driven) triage approach may be identified: risk stratification ability, scalability, continuous integration of newly acquired knowledge and accuracy. Indeed, ML models can be trained to predict outcome events such as mortality, hospitalization, or readmission [40, 41]. This can help prioritizing patients who need urgent medical attention and ensure that resources are optimally allocated. Regarding scalability, one may easily imagine a situation of increasing demand for medical triage due to pandemics, natural disasters, or mass casualty incidents, where ML models can help scale up triage assessments and adapt triage thresholds to available resources quickly and accurately. Facing a newly discovered disease such as COVID-19, medical knowledge is being progressively built while taking care of new patients. Through near real time analysis of incoming patient data, ML based adaptive triage can seeminglessly incorporate this newly acquired knowledge into the decision-making process. Finally, ML may also benefit to the triage process itself. High triage accuracy is expected

as ML models can process large amounts of data from various sources, such as electronic health records, vital signs, lab tests, and patient history, to make more accurate triage assessments. Human error may also be reduced using these automated processes.

The drawback of a ML approach is the requirement of an immediately available and as comprehensive as possible dataset regarding medical records, patients flow and ICU capacity. At the time of first COVID-19 outbreak in 2020, part of the medical community in France decided that the exponential kinetics of the outbreak warranted the quick development of a triage tool. But even if ICU available beds and COVID-19 positive testing flow were carefully monitored, severe patients flow and comprehensive medical data were barely available. With such limited data availability, ML-based triage was not applicable. The SFAR-SSA triage strategy was thus developed a priori based mostly on short track literature, and we later performed this a posteriori validation study from retrospective data. We believe that ML based triage for a fast spreading outbreak of a newly discovered disease might be applicable only with pre-developed ML algorithms in a healthcare system with highly centralized, near real time data collection, hence with a strong requirement for strict data privacy protection. There have been few actual examples of this approach in the first COVID-19 outbreak, if any. However, the retrospective validation of ML based COVID-19 triage systems could help preparing for future pandemics [42].

### Triage algorithms and ethical guidelines

Of note, the SFAR/SSA triage protocol does not contend against ethical recommendations. On the contrary, in an extreme situation only, it does support them and helps comply with them by providing a formal framework for fair decision-making under such strain [17].

### Conclusion

This study retrospectively validates the early developed SFAR/SSA critical care triage protocol in case of COVID-19 related saturation of ICU resources without significant population immunity, to be used as a last line strategy when even ICU capacity extension and patient transfers can no longer meet critical care needs. In such situations, this protocol would enable fair resource allocation, thereby limiting avoidable deaths and maximizing the number of lives saved, in compliance with highest ethical standards.

SARS-CoV-2 has been severely impacting healthcare systems since its emergence. In many countries, besides transmission control measures, the unexpectedly quick availability of efficient vaccines, the efficacy of mass vaccination campaigns and the genomic evolution of circulating viral variants have dramatically reduced the strain on healthcare systems and especially on ICUs in many areas. The need for triage has thus become less obvious. However, the sudden opening of vast areas previously under "zero COVID" policies with unknown actual population immunity causes a massive viral circulation. This might lead to the emergence of new variants poorly covered by previous natural or vaccine immunity, hence to new saturating epidemic waves. This validated triage protocol could provide valuable help to authorities and physicians dealing with such new surges of critically ill COVID-19 patients. Its integration into the armamentarium against COVID-19 is thus still warranted.

Interestingly, the COVID-19 crisis has also shown the reluctance to merely consider triage, both from many physicians not trained in disaster medicine and from healthcare policy decision-makers, not to mention from populations used to rely on high level healthcare systems. In order to prepare for potential similar crises with a risk of overwhelmed treatment capacity in the future, the present clinical validation of the SFAR/SSA critical care triage protocol is highly valuable to help explain and teach triage and make it acceptable as a coherent, ethical and helpful medical strategy, although to be used only in extreme situations.

## Supporting information

**S1 Appendix. Simulated triage and quantification of lives potentially saved by triage.**
Detailed description of how a simulated triage cohort was generated and used to estimate the
number of supplementary lives potentially saved by triage as compared with absence of triage,
in a situation of saturated critical care capacities in a COVID-19 overwhelming outbreak.
(PDF)

**S1 Table. Patient severity and outcome by initial priority level (day 0) in tension.** Patient
severity and outcome according to the priority level assigned at ICU admission (day 0, first
step of the SFAR/SSA critical care triage protocol) in a situation of tension in critical care
capacities.
(PDF)

**S2 Table. Comparison of priority levels between both triage steps in tension.** Since no
recovered cardiac arrest was recorded during initial ICU stay, the second step of priority allo-
cation (on day 7 to 10) in tension was identical to that in saturation.
(PDF)

**S1 Fig. Outcome of COVID-19 ICU patients by initial priority level (day 0) in tension.**
Cumulative incidence (c.i.) of alive discharge from ICU and survival (= 1 –c.i. of death in
ICU) for COVID-19 patients. A: P4 compared with other priority levels at day 0. B: compari-
son between all priority levels at day 0. Shaded areas: initial prioritization no longer relevant
due to reassessment. Since no recovered cardiac arrest was recorded during initial ICU stay,
the second step of priority allocation (on day 7 to 10) in tension was identical to that in satu-
ration.
(PDF)

**S2 Fig. Resource utilization by initial priority level (day 0) in tension.** Length of ICU stay
(A) and length of mechanical ventilation (B) by initial priority level (day 0) in tension.
(PDF)

**S3 Fig. SAPS2 (A) and age (B) distributions by initial priority level (day 0) in tension.**
(PDF)

**S1 Data.**
(CSV)

**S1 File.**
(PDF)

## Author Contributions

**Conceptualization:** Nicolas Donat, Benoît Veber, Thomas Leclerc.

**Data curation:** Nicolas Donat, Audrey Cirodde, Sébastien Gette, Pierre Gildas Guitard, Clém-
ent Hoffmann.

**Formal analysis:** Nicolas Donat, Thomas Leclerc.

**Investigation:** Nicolas Donat, Nouchan Mellati, Thibault Frumento, Audrey Cirodde, Sébas-
tien Gette, Pierre Gildas Guitard, Clément Hoffmann.

**Methodology:** Thomas Leclerc.

**Supervision:** Benoît Veber, Thomas Leclerc.

**Validation:** Nicolas Donat, Benoît Veber, Thomas Leclerc.

**Visualization:** Thomas Leclerc.

**Writing – original draft:** Nicolas Donat.

**Writing – review & editing:** Nouchan Mellati, Thibault Frumento, Benoît Veber, Thomas Leclerc.

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
