## [Decision Letter · Decision Letter 0]

27 Mar 2023

PONE-D-22-35651Validation of a pre-established triage protocol for critically ill patients in a COVID-19 outbreak under resource scarcity: a retrospective multicenter cohort studyPLOS ONE

Dear Dr. Donat,

Thank you for submitting your manuscript to PLOS ONE. After careful consideration, we feel that it has merit but does not fully meet PLOS ONE’s publication criteria as it currently stands. Therefore, we invite you to submit a revised version of the manuscript that addresses the points raised during the review process.

We look forward to receiving your revised manuscript.

Kind regards,

Stefan Grosek, Ph.D., M.D.,

Academic Editor

PLOS ONE

Journal Requirements:

Additional Editor Comments:

Dear Authors

Thank you for the manuscript submitted to the PLOS One. Both reviewers and myself found your manuscript well written methodologically and scientifically, however the 2nd reviewer suggested to discuss the difference beteewen your approach and a data driven approach (ML).

Please discuss it and send it back for review.

Kind regards

Academic Editor

Reviewers' comments:

Reviewer's Responses to Questions

**Comments to the Author**

1. Is the manuscript technically sound, and do the data support the conclusions?

Reviewer #1: Yes

Reviewer #2: Yes

2. Has the statistical analysis been performed appropriately and rigorously? 

Reviewer #1: Yes

Reviewer #2: Yes

3. Have the authors made all data underlying the findings in their manuscript fully available?

Reviewer #1: Yes

Reviewer #2: Yes

4. Is the manuscript presented in an intelligible fashion and written in standard English?

Reviewer #1: Yes

Reviewer #2: Yes

5. Review Comments to the Author

Reviewer #1: The manuscript provides efficiency analysis of early French triage algorithm potentially used in scarcity of critical care resources during COVID 19 epidemics. The analysis is done on retrospective data, and analysis is divided into two scenarios- during saturation (situation without additional ICU resources) or during less severe situation- tension.

The algorithm categorized critically ill patients by probability of survival based on medical history and severity, with four priority levels for initiation or continuation of critical care: P1 – high priority, P2 –intermediate priority, P3 – not needed, P4 – not appropriate.

Manuscript analysis saturation scenario, in the appendix there is also analysis of tension.

Priority levels were retrospectively allocated at ICU admission and on ICU day 7-10. Mortality rate, cumulative incidence of death and of alive ICU discharge, length of ICU stay and of mechanical ventilation were compared between priority levels. Calculated mortality and survival were compared between full simulated triage and no triage.

Simulation under saturation showed that this two-step triage protocol could have saved more lives than no triage. Although it cannot eliminate potentially avoidable deaths, this triage protocol proved able to adequately prioritize critical care for patients with highest probability of survival, hence to save more lives if applied.

The problematic describe is the manuscript is actual, the manuscript is very well written, conclusion are based on the data. The ethical dilemmas and weak points are well addressed.

I have no additional comments.

Reviewer #2: Dear Authors, many thank for the opportunity to read your paper: it seems very good paper,

I do not any concerns about your work, methodology is very innovative: it could be interesting discussing better the difference beteewen your approach and a data driven approach (ML)

6. PLOS authors have the option to publish the peer review history of their article (what does this mean?). If published, this will include your full peer review and any attached files.

Reviewer #1: No

Reviewer #2: No

---

## [Author Response · Author response to Decision Letter 0]

23 Apr 2023

Dear Editor, Dear reviewers,

We implemented in our manuscript a discussion about the potential benefits of machine learning approach for triage algorithms. The paragraph is called « Triage algorithms and Machine Learning » and is located in the discussion part between the « study limitations » part and the « triage algorithms and ethical guidelines » part.

As requested, we made sure that the manuscript meets PLOS ONE's style requirements. References were also verified.

Best Regards,

Nicolas DONAT

---

## [Editor Report · Decision Letter 1]

28 Apr 2023

Validation of a pre-established triage protocol for critically ill patients in a COVID-19 outbreak under resource scarcity: a retrospective multicenter cohort study

PONE-D-22-35651R1

Dear Dr. Donat,

We’re pleased to inform you that your manuscript has been judged scientifically suitable for publication and will be formally accepted for publication once it meets all outstanding technical requirements.

Kind regards,

Stefan Grosek, Ph.D., M.D.,

Academic Editor

PLOS ONE

Additional Editor Comments (optional):

Dear Authors

Thank you very much for addressing all issues recommended by the 2nd reviewer. 

Kind regards, Stefan Grosek, academic editor
---

## [Editor Report · Acceptance letter]

4 May 2023

PONE-D-22-35651R1 

Validation of a pre-established triage protocol for critically ill patients in a COVID-19 outbreak under resource scarcity : a retrospective multicenter cohort study 

Dear Dr. Donat:

I'm pleased to inform you that your manuscript has been deemed suitable for publication in PLOS ONE. Congratulations! Your manuscript is now with our production department. 

Kind regards, 

on behalf of

Professor Stefan Grosek 

Academic Editor

PLOS ONE